# Chronic Exposure to Chlorpyrifos Damages Thyroid Activity and Imbalances Hepatic Thyroid Hormones Signaling and Glucose Metabolism: Dependency of T3-FOXO1 Axis by Hyperglycemia

**DOI:** 10.3390/ijms24119582

**Published:** 2023-05-31

**Authors:** Teresa Peluso, Valeria Nittoli, Carla Reale, Immacolata Porreca, Filomena Russo, Luca Roberto, Antonia Giacco, Elena Silvestri, Massimo Mallardo, Mario De Felice, Concetta Ambrosino

**Affiliations:** 1Department of Science and Technology, University of Sannio, Via de Sanctis, 82100 Benevento, Italy; 2Biogem Scarl, Institute of Molecular Biology and Genetics Research, Via Camporeale, 83031 Ariano Irpino, Italy; 3Department of Molecular Medicine and Medical Biotechnologies, University of Naples “Federico II”, Via Pansini 5, 80131 Naples, Italy; massimo.mallardo@unina.it; 4Institute of Experimental Endocrinology and Oncology (IEOS), CNR, Via Pansini 6, 80131 Naples, Italy

**Keywords:** thyroid hormones, endocrine disruptor, glucose metabolism, deiodinases

## Abstract

Early life exposure to Endocrine Disruptor Chemicals (EDCs), such as the organophosphate pesticide Chlorpyrifos (CPF), affects the thyroid activity and dependent process, including the glucose metabolism. The damage of thyroid hormones (THs) as a mechanism of action of CPF is underestimated because the studies rarely consider that TH levels and signaling are customized peripherally. Here, we investigated the impairment of metabolism/signaling of THs and lipid/glucose metabolism in the livers of 6-month-old mice, developmentally and lifelong exposed to 0.1, 1, and 10 mg/kg/die CPF (F1) and their offspring similarly exposed (F2), analyzing the levels of transcripts of the enzymes involved in the metabolism of T3 (*Dio1*), lipids (*Fasn*, *Acc1*), and glucose (*G6pase*, *Pck1*). Both processes were altered only in F2 males, affected by hypothyroidism and by a systemic hyperglycemia linked to the activation of gluconeogenesis in mice exposed to 1 and 10 mg/kg/die CPF. Interestingly, we observed an increase in active FOXO1 protein due to a decrease in AKT phosphorylation, despite insulin signaling activation. Experiments in vitro revealed that chronic exposure to CPF affected glucose metabolism via the direct modulation of FOXO1 activity and T3 levels in hepatic cells. In conclusion, we described different sex and intergenerational effects of CPF exposure on the hepatic homeostasis of THs, their signaling, and, finally, glucose metabolism. The data points to FOXO1-T3-glucose signaling as a target of CPF in liver.

## 1. Introduction

The disarrangement of different signaling pathways dependent on steroid hormones, including thyroid hormones (THs), is involved in the development and progression of several diseases, including obesity, insulin resistance, and hyperlipidemia. These are components of the metabolic syndrome that is becoming a global issue because of the spreading of the adoption of unhealthy lifestyles and of the environmental Endocrine Disruptors Compounds (EDC) [1]. EDCs include a wide variety of chemicals (phathalates, bisphenols, etc.) used in the plastics industry and in consumer products [2,3], as well as compounds used in farming as pesticides [2]. They all can exhibit health-threatening effects at low doses (below those used in traditional toxicological studies) that may not be predicted by the effects at higher doses [4,5]. Chlorinated organophosphate pesticides (OPs), including Chlorpyifos (CPF), are raising a big concern because of their detection at concentrations ranging from 0.002 to 436 μg/kg in a wide range of foods, not only fruits and vegetables but also meat, fish, and eggs [6]. These doses are considerably high, although CPF is quickly inactivated [7,8,9,10]. Considering that exposure to CPF may cause serious toxic effects in exposed humans and animals, the US Environmental Protection Agency (EPA) imposed a ban on the sale of CPF for residential use in December 2001 [11]. In Jannuary 2020, the European Union (EU) commission found it appropriate not to renew the approval of the CPF. However, its use was continued for controlling the crop damage from insects [12]. Several studies have investigated their effects on the high-risk behavior or sleeping in young adults living in proximity to agriculture areas (CHAMACOS study and NHANES study, respectively) [13]. Exposure to pesticides has been associated with metabolic disease for some time [14]. Sub-chronic exposures to CPF have been associated with incidence of diabetes mellitus via a mechanism implying the damage of insulin and TH signaling [15]. Indeed, hepatic gluconeogenesis and glycogenolysis in response to insulin [16] and thyroid hormones are dysregulated in type 1 and 2 diabetes mellitus [17]. The metabolism and signaling of THs can be deeply impacted by EDCs mediating, as well as various metabolic disorders [18,19]. Experimental studies evidenced this role of EDCs, above all when the exposure starts in utero or in neonatal stages [20,21,22]. The high vulnerability to EDC exposure in the early life stages is linked to the reduced activity of detoxifying pathways [23]. Indeed, we have shown in zebrafish that developmental and lifelong exposure to CPF exert different intergeneration effects on thyroid and liver health [24]. However, the thyroid-disrupting effects of CPF have only recently started to be studied.

Since early life exposure to EDCs has been associated with an increasing risk of endocrine and metabolic disease in adulthood and in subsequent generations [25], we decided to investigate the effects of different doses of CPF in rodents born from lifelong-exposed individuals and that are developmentally and lifelong exposed (Appendix A) in order to reproduce a context of exposure more relevant to humans. By using the RT-qPCR analysis, we investigated the expression levels of genes involved in THs metabolism and signaling (*Thr* and *Dio1*) and glucose (*G6pase* and *Pck1*) and lipid metabolisms (*Fasn* and *Acc1*). We focused on TH-specific targets in the liver, looking at FOXO1 and at insulin signaling pathways. We pointed to peripheral disarrangement of TH signaling and of its crosstalk with other pathways involved in hepatic control of the glucose metabolism as a mechanism of action of CPF.

## 2. Results

### 2.1. Developmental and 6-Month Exposure to CPF of Offspring of Dams Similarly Exposed Impair Thyroid and Metabolic Health Only of the Former in Sex-Dependent Manner

We and others have reported that developmental exposure to CPF promoted thyroid dysfunctions [15]. Indeed, 6-month-old females exposed since their conception and lifelong to 0.1, 1, and 10 mg/kg/die CPF (F1 generation, F1 from now on, exposed as reported in Appendix A) were euthyroid. We determined the circulating level of free thyroxine (T4) in exposed males of the same litter by ELISA assay (cfT4, from now). No statistical difference was detected, as in the females (Appendix A). Moreover, we observed that exposed F1 males and females did not show any significant difference in body weight in all the used conditions, compared to the control (Appendix A). In order to assess if the developmental exposure to CPF of the mothers (F1) could affect the thyroid health of the second generation (F2) mice, which were grown in the same environmental condition (schematized in Appendix A), we determined the cfT4 levels by ELISA assay in this generation. As shown in Figure 1A, a trend toward the decrease was evidenced in males exposed to the different doses of CPF (0.1, 1 and 10 mg/kg/die, respectively, lower, milder, and higher doses, from now on). On the contrary, cfT4 increased only at CPF 1 mg/kg/die exposed females (Figure 1B).

Hypothyroidism has been associated with the increase of body weight. Therefore, we monitored this parameter in F2 males and females (Figure 2A and Appendix A, respectively). Chronic exposure to CPF promoted weight gain in all exposed F2 males (+17.16%, +4%, and +1.47% vs. CRTL), with the lower dose having the greatest effect (Figure 2A). No major effect was detected in F2 females (Appendix A).

Then, we determined the level of fasting glucose in the exposed F2 mice. In CPF-exposed F2 males, we detected a statistically significant increase in fasting glucose in mice treated with 10 mg/kg/die (+43.13% vs. CRTL, Figure 2B), whereas its reduction was detected in F2-exposed females at the same dose (Appendix A). We investigated the glucose tolerance in F2 animals by using the Oral Glucose Tolerance Test (OGTT). We observed a statistically significant increase in glucose level at 90 min after its administration in mice treated with the lower dose of CPF and a trend toward the increase in mice exposed to the higher dose (Figure 2C,E). In mice treated with a mild dose of CPF, we did not find a significant difference compared with the control (Figure 2D). No major impact was detected in 6-month F2 females (Appendix A).

### 2.2. CPF Weakly Affects Hepatic Lipid Metabolism in Hypothyroid-Exposed F2 Mice Males

The thyroid state significantly affects various stages of lipid metabolism, including hepatic lipolysis and de novo lipogenesis [26,27,28,29]. Thus, we firstly analyzed the levels of transcripts of key enzymes involved in lipid metabolism, such as the fatty acid synthase (*Fasn*) and acetyl-CoA carboxylase (*Acc1*), by RT-qPCR. We discovered that only the former was reduced in F2 males exposed to mild and high doses of CPF (Figure 3A,B). We did not detect any major difference in blood triglycerides among the different exposure conditions, with the exception of lower doses of CPF (Figure 3C), while we observed a trend toward the increase in the levels of cholesterol at the high CPF doses (Figure 3D) [30]. 

### 2.3. CPF Alters the Insulin Response in Exposed F2 Livers

The liver also plays a fundamental role in coordinating glucose metabolism. Both THs and insulin act on hepatocytes to regulate glucose homeostasis via multiple mechanisms. Thus, we investigated the insulin signaling in liver of the F2 CPF-exposed males. An increase of the p-IRTyr972 level was detected in mice treated with 1 and 10 mg/kg/die (Figure 4A,B). We also analyzed the levels and the phosphorylation of the IRS1 protein, an important player in controlling insulin receptor signaling and in promoting insulin resistance [31,32]. We found a trend toward the increase of p-IRS1 Ser302 in 1 mg/kg/die and 10 mg/kg/die in liver from exposed F2 males, whereas it was decreased in 0.1 mg/kg/die samples (Figure 4A,C). Concordantly with the role of Ser/Thr phosphorylation in the modulation of IRS1 degradation by proteasome [30], we retrieved the level of p-IRS1Ser302 protein increased only in the latter (Figure 4A,C). Insulin engagement of its cognate receptor leads to AKT phosphorylation and activation. We observed an increase of p-AKT Ser473 and consequently of p-GS3KSer21 in mice treated with CPF 0.1 mg/kg/die (Figure 4D–F), but not in 1 and 10 mg/kg/die exposed livers, in which we detected the reduction of both signals (Figure 4D–F), suggesting an unresponsiveness to insulin signaling of livers exposed to milder and higher doses of CPF.

### 2.4. CPF Modulate Hepatic Glucose Homeostasis by Acting on Local T3-FOXO1 Axis

Consistently with the hyperglycemia (Figure 2B) and with the inactivation of AKT signaling at milder and higher doses CPF (Figure 4D–F), we investigated the expression of two of main important transcripts modulating the production of glucose by the liver, the glucose-6-phosphatase (*G6pase*), and the phosphoenolpyruvate carboxykinase 1 (*Pck1*). In particular, *Pck1* controls the hepatic gluconeogenesis, while *G6pase* is involved in gluconeogenesis and glycogenolysis, catalyzing the final step of both processes. Concordant with previous results, we found a normal expression of both transcripts in livers from mice exposed to low-dose CPF (Figure 5A,B), showing also an increase of AKT phosphorylation (Figure 4D,E). We observed an increase in the *G6pase* and *Pck1* mRNAs in the liver of F2 males exposed to 1 and 10 mg/kg/die CPF (Figure 5A,B). Interestingly, *Glut2* (*Slc2A2*) mRNA (Figure 5C), which codifies for the transporter involved in the glucose efflux from liver, was found to be increased in mice exposed to a higher dose of CPF.

Forkhead transcription factor, FoxO1, is highly expressed in insulin-responsive liver, in which it orchestrates the transcriptional cascade that regulates glucose metabolism [33], including *Pck1* and *G6pase* mRNAs’ modulation [34,35]. Specifically, insulin suppresses FoxO1 activity through activation of the PI3K/AKT signaling pathway [36].

Therefore, we assessed the FOXO1a protein level and phosphorylation in our samples. We evaluated by Western blotting the hepatic level of p-FOXO1a Ser256 and FOXO1a proteins (Figure 6A). As shown in Figure 6C, FOXO1a protein level was increased in all the exposure groups, whereas no major difference was retrieved in its phosphorylation (Figure 6B).

It was reported that TH integrates cell metabolic status controlling *Foxo1a* transcription factor activity. Indeed, by inhibiting the AKT signaling, T3 promotes the permanence of *Foxo1a* in the nuclei and modulates the transcription of genes related to glucose metabolism [36]. Considering the hypothyroid status of exposed F2 males and that the free triiodothyronine (free T3 is fT3 from now on) intra-organ levels are mainly established peripherally, we evaluated the hepatic effect of CPF on TH homeostasis, detecting hepatic levels of fT3 (hfT3 from now on) and fT4 (hfT4 from now on) via ELISA assay. We used as samples the hepatic lysates prepared, as reported in the M&M section. As shown in Figure 6E, a dose-dependent decrement of the hfT3 was detected despite the fact that no major decrease in hfT4 was evidenced (Figure 6D). The data were corroborated by the increase of *Dio1* mRNA, codifying for the enzyme promoting fT4 conversion in fT3 in the adult liver (Figure 6F). To verify the activation of the feedback loop, which is active along the HPT axis, we analyzed the *Tsh* mRNA in the pituitary. It was increased in the pituitary of all the exposed males, especially in mice exposed to a low dose of CPF (Appendix A). Furthermore, we measured the level of the transcripts of the TH receptors *Thra* and *Thrb.* A trend toward the increase was evidenced for *Thra* and *Thrb* mRNAs, reaching statistical significance in 1 mg/kg/die CPF livers for the former and 0.1 mg/kg/die CPF livers for the latter (Appendix A).

### 2.5. Higher Doses of CPF Mimic the Presence of T3 Leading to Hyperglycemia

The obtained results revealed that CPF affects liver glucose metabolism, acting on FOXO1a protein activity and influencing local T3 homeostasis. Thus, we set up an in vitro experiment to better investigate CPF action. We used the human hepatic carcinoma cells line, HepG2, cultured in normal medium (DMEM 1 g/L glucose, NG from now on,) or in high-glucose condition (DMEM 5 g/L, HG from now on) and exposed them to different doses of CPF, as described in M&M section. We first observed CPF effects on mRNAs’ levels of glucose-metabolism-related marker genes, such as *G6pase* and *Pck1*. As reported in Figure 7A,B, in the normal glucose condition, CPF upregulated the expression of *G6pase* transcript at all tested doses, while *Pck1* mRNA was found increased in a statistical significant manner only in higher doses of CPF. Conversely, HepG2 cells grown in a high-glucose medium showed a reduction of both transcripts when cultured with low doses of CPF, while an increase of their expression was detected in the higher-dose CPF (Figure 7C,D). The obtained results suggested that a high CPF dose stimulated glucose metabolism processes, leading to inability of hepatocytes to sense the glucose increase, when the cells were grown in high external glucose condition.

Considering that the expression of both analyzed genes was dependent on FOXO1a activity and that we have found an increase of FOXO1a protein in exposed F2 livers, we investigated the expression of total and phosphorylated form of FOXO1a protein in HepG2 cells. We found that, in all different doses of CPF used and in NG (Figure 8A,B), the phosphorylation of FOXO1a was decreased, while the total form of FOXO1a was increased (Figure 8A,C), suggesting that CPF was able to stimulate glucose metabolism in hepatocytes. In the high-glucose condition, we observed a statistically significant increase of FOXO1A phosphorylation, which was more pronounced in cells cultured with low doses of CPF (Figure 8D,E), together with a strong upregulation of total FOXO1a protein in the same samples. Concordantly, in HepG2 cells grown in low doses CPF and in HG, the increase of FOXO1a phosphorylation was related to the reduction of *G6pase* and *Pck1* mRNAs (Figure 7C,D). Differently, in HepG2 cells grown in HG and treated with the higher dose of CPF, we detected a trend toward the increase of total FOXO1a protein (Figure 8D,F) and, despite the increase of FOXO1a phosphorylation, both the expression of *G6pase* and *Pck1* was found upregulated (Figure 7C,D).

These obtained results prompted us to test the effects of different doses of CPF on intracellular levels of fT3 in HepG2 cells, considering that T3 might regulate FOXO1-mediated transcription.

In normal glucose, the increase of the total FOXO1a protein and the statistically significant upregulation of *G6pase* mRNA (Figure 7A) was in agreement with the increase of fT3 level detected by ELISA assay in cellular proteins (Figure 8G). These results suggested that CPF alone influences the T3 levels and, consequently, may modulate the T3-FOXO1a axis, leading to gene transcription. However, the trend toward the increase of fT3 level found in higher doses of CPF in NG condition did not justify the activity of the FOXO1a protein and the strong upregulation of *G6pase* and *Pck1* mRNA in these samples (Figure 7A,B). These results suggested that higher doses of CPF produced a more pronounced activity of FOXO1a protein, even without a strong impact on T3 levels.

Conversely, in HG, CPF generated a statistically significant reduction of fT3 in HepG2 cells treated with low-dose CPF (Figure 8H), while a trend toward the decrease of fT3 levels was detected in HepG2 cells grown in high CPF doses (Figure 8H). Thus, the fT3 levels reflected the expression of glucose-metabolism-related marker genes in both NG and HG conditions, with the exception for the higher dose of CPF. Indeed, HepG2 cells cultured with low doses of CPF were still able to regulate glucose metabolism. On the contrary, HepG2 cells cultured with the higher dose of CPF were unable to block the processes leading to glucose production, even when the external glucose was high. In this condition, the activation of both gluconeogenesis and glycogenolysis processes may lead to the establishment of hyperglycemia.

In conclusion, the in vitro data suggest that CPF might mimic the T3 presence inside the cells and that this could be the possible mechanism contributing to the insulin resistance.

## 3. Discussion

Epidemiological and experimental data suggest that early life exposure to EDCs, including CPF, impacts endocrine and metabolic health at the adulthood [11,37,38,39]. Considering the complexity of the human exposome, experimental studies conducted in animal models are pivotal in defining the effects and the mechanisms of action of different compounds and, as a consequence, the risks associated to their exposure for humans. In this work, we attempted to reproduce, in the laboratory, an exposure scenario relevant to human in terms of exposure routes, exposure windows, and doses. In particular, we investigated the effects of CPF exposure, focusing on animals developmentally and lifelong exposed (F2), that originates from exposed parents (F1). Previous data from the literature indicate that CPF exposure principally affects the nervous systems [40]. Recently, we and others evidenced that developmental and lifelong exposure to CPF promoted dysthyroidisms, even in evolutionarily distant species, such as rodents and zebrafish [15,24]. In particular, our results obtained in zebrafish suggested that parents’ developmental and lifelong exposure to CPF presented signs of dysthyroidisms and of an imbalanced hepatic metabolic activity weaker than the ones developed by their offspring, who were similarly exposed [24]. These data are corroborated by the results in F2 mice here reported, revealing their stronger dysthyroidism than in F1. Noteworthy, what owe bserved in zebrafish ruled out any role of maternal thyroid dysfunction in altering the endocrine and metabolic health because, in zebrafish, the embryos develop externally. Overall, this suggests that, in evolutionarily distant vertebrate parents, exposure to CPF “sensitizes” the endocrine health of the offspring. The similarity of the effects in evolutionarily distant vertebrate models is suggested as a valid parameter to identify a conserved alteration that is verifiable in humans. Several studies have described cognitive and behavior alteration in the kids, adolescents, and adults exposed to CPF in early life stages [13]. Studies conducted on farmers using pesticides or living in agricultural areas confirmed the data on other workers [41,42]. However, few studies have involved their progeny without evidencing any main effects [43]. This might depend on the time, anyway, required to develop the phenotype due to long lifespans and the relative smaller number of offspring in humans. Nevertheless, our results suggest a risk for developmentally exposed individuals of developing endocrine/metabolic diseases later in the life or subsequent to exposure to other environmental cues. These lasts ones might be represented by the alteration of other signaling pathways linked or not to the exposure. It is well-known that THs regulate glucose metabolism and homeostasis. Thus, the dysregulation in serum and intra-tissue content of THs can compromise metabolic pathways in liver and might promote hyperglycemia and hyperinsulinemia [44]. As in zebrafish liver, we showed that, in murine liver, the hepatic metabolism of THs is targeted by CPF. Indeed, we found an upregulation of deiodinase 1 (*Dio1*) mRNA, concomitantly with a decrease of liver fT3 level, suggesting a condition of hypothyroidism in CPF exposed F2 mouse livers. Moreover, these mice were also affected by hyperinsulinemia and hyperglycemia. FOXO proteins are key regulators of metabolic health and functions [45]. In particular, FOXO1 contributes to maintain the balance between gluconeogenesis and glycogenolysis [40]. FOXO1 is phosphorylated by AKT, activated by insulin, that promotes its exclusion from the nucleus and inactivates its transcriptional activity, leading to the inhibition of target genes’ transcription, such as *Pck1* and *G6pase* genes [46]. Interestingly, we found that the total form of FOXO1a protein was increased in exposed F2 male livers compared with the control, particularly in the case of mild and high doses of CPF. This result is corroborated by the increase, in both exposure groups, of *G6pase* and *Pck1* mRNAs’ direct targets of *Foxo1a*. This leads to glucose metabolism activation that, together with the increase of *Glut2* transcript, sustained the systemic hyperglycemia. On the contrary, AKT phosphorylation was found to be increased only in mice treated with a low dosage CPF, together with the phosphorylation of GS3K, indicating that insulin signaling was still working. Thus, in F2 livers exposed to 0.1 mg/kg/die CPF, the upregulation of SBREP1c, together with the increase in body weight and in levels of triglycerides, may be the result of the activation of lipogenesis process [47]. Conversely, chronic exposure to 1 and 10 mg/kg/die CPF dysregulated signaling downstream the IR, including FOXO1a signaling. Among the involved pathway in the regulation of FOXO1, hepatic thyroid hormones are key players. Indeed, T3 increase FOXO1 nuclear localization, DNA binding, and target gene transcription by reducing AKT-dependent FOXO1 phosphorylation in a THRB1-dependent manner [36]. The hypothyroid status of the liver was not consistent with the increase of FOXO1A in all the exposed samples. However, in CPF 0.1 mg/kg/die livers, the FOXO1a responsive genes, such as *G6pase* and *Pck1*, were not induced, mainly because of the activation of AKT signaling. On the contrary, in livers exposed to 1 mg/kg/die and 10 mg/kg/die CPF, we detected the increased phosphorylation of IR without AKT signaling activation. The last result points to the reduction of hepatic T3 as the factor contributing to the imbalance PI 3-kinase→Akt→FOXO1 cascade by upregulation of the FOXO1 protein levels. This activity is enforced at the higher dose of CPF since the increased levels of p-IRS1 Ser 302 phosphorylation, usually part of the physiological feedback mechanism involved in silencing the insulin receptor signaling deregulated, indeed, in insulin resistance. Different Ser/Thr kinases can target this site, whose activation can be AKT dependent or independent [48]. The increased phosphorylation of AKT only at the low dose was in agreement with the observation that the IRS1 protein levels were increased only in this condition. At higher doses, the levels of IRS1 were unchanged, but we detected the increase of the inhibitory Ser/Thr phosphorylation. These events are controlled by different kinases, including PI3K and S6K, that are active in the physiological inhibition of IR signaling [48]. Both kinases are activated by T3 in hepatocytes and in other cellular contexts [49]. Furthermore, T3 positively modulates the activity of S6 kinase [50]. Overall, the increase of IRS1 phosphorylation of Ser302 is not in agreement with the detected hypothyroidism. Of course, other different factors may contribute to these effects. It is noteworthy that inflammatory cytokines promote insulin resistance, increasing the Ser302 phosphorylation of IRS1 [42,51]. At the same time, the results obtained in vitro suggest a direct role of CPF in regulating the T3-FOXO1 axis to induce altered glucose metabolism. Indeed, at lower doses of CPF, hepatic cells’ response is dependent on glycemia. Thus, in the hyperglycemic condition, HepG2 cells exposed to low-dose CPF were still able to block glucose metabolic processes, as indicated by the decrease of both genes involved in gluconeogenesis and glycogenolysis, together with fT3 level (Figure 7C,D and Figure 8H). However, at a high dose of CPF, the HepG2 cells were not able to discriminate their environmental growth conditions, leading to an increase in glucose metabolism by upregulation of *G6pase* and *Pck1* mRNAs’ expression, despite a quite normal level of fT3 (Figure 8H). Finally, we observed that chronic exposure to CPF, especially in high doses, might alter the liver FOXO1-T3-glucose axis, a mechanism that probably leads to insulin resistance. Overall, the data here reported suggest that the hepatic hypothyroidism detected in F2-exposed males could simulate a fasting condition although insulin signaling is activated. Indeed, beyond the hepatic low level of T3, CPF induces a FOXO1a nuclear increase that results in *G6pase and Pck1* activation. In this context, the increased phosphorylation of the inhibitory sites of IRS1 might represent a factor further exacerbating the insulin resistance. Mechanistically, our data evidence how environmentally promoted impairment of TH secretion and of their peripheral metabolism and signaling plays an important role in unbalancing the insulin signaling and originating the “pandemic insulin resistance and hyperglycemia”. Furthermore, we point out the necessity to conduct intergenerational studies in scenarios that are environmentally relevant in order to proper evaluate the risks related to exposure to EDCs because of the increased susceptibility to develop diseases in different generations.

## 4. Materials and Methods

### 4.1. Animal Model

Animal experiments were performed in accordance with the European Council Directive 86/609/EEC, following the rules of the D.Lgs 116/92 (ID number 25–10). Procedures were approved by the Ethical committee, named CESA (Committee for the Ethics of the Experimentations on Animals), of the Biogem S.c.ar.l. Mice were kept under standard facility conditions and received water and a standard diet (4RF21, Mucedola, Settimo Milanese, Italy) ad libitum. CD1 dams (outbred strain, 8 mice/treatment group) were exposed as previously described [14]. Therefore, the offspring were exposed through the mothers from gestational day 0 (GD0) till the weaning. Then, the offspring (10 females and males) were directly exposed. The doses of pesticide were chosen up to published reports in which no systemic effects of the exposure were reported, consistent with the NOAEL for CPF (0.3 mg/kg/die) and its relevant long-term NOAEL (0.9 mg/kg/die, 18 months) reported in the last statement of EFSA for CPF (approved 31 July 2019) [52]. Animals were sacrificed at 6 months for blood and organ collection.

### 4.2. Cell Line

Human HepG2 hepatic carcinoma cell line was obtained from the American Type Culture Collection (ATCC). HepG2 cells were cultured in Dulbecco’s modified Eagle’s medium (DMEM, Gibco, Grand Island, NY, USA) supplemented with 10% fetal bovine serum (Sigma, Burlington, MA, USA), penicillin (100 units/mL, Gibco), and streptomycin (100 μg/mL, Gibco) and were maintained in a humidified incubator under 5% CO2 at 37 °C. For the high-glucose condition, HepG2 cells were cultured in DMEM (High Glucose). HepG2 cells were seeded at equal densities directly into wells of a standard 6-well plate (Nunc plate) and exposed to 10^−5^ M, 10^−7^ M, 10^−8^ M, and 10^−9^ M CPF for 7 days of treatment. Normal or high-glucose growth medium with each different dose of CPF was changed every 2 days.

### 4.3. Glycemia Measurement

Blood glucose levels were monitored from the tail tip, using a handheld glucometer (Contour XT, Bayer, Milan, Italy) in the basal state.

### 4.4. OGTT Test

Mice were fasted overnight (16–18 h), with free access to water. Mice were anesthetized before oral gavage of glucose, 50% *w*/*v* solution, in a dose corresponding to 2 g glucose per kg bodyweight. Glycemia was measured after tail-tip punctures, at times 0, 30, 60, 90, and 120 min after oral glucose load, using a handheld plasma calibrated glucometer (Accu-check compact plus, F.Hoffmann-La Roche AG, Bazel, Switzerland).

### 4.5. Hormone Measurements

Serum and HepG2 cells were centrifugated at 5000× *g* for 5 min at 4 °C. The cfT4 and cfT3 levels were determined by a specific ELISA test (fT4 and fT3 ELISA kit were from Diametra, Perugia, Italy), according to the manufacturer instructions. Intra-tissue hepatic levels of fT4 and fT3 levels were measured on liver homogenates (*n* = 3/group) or HepG2 cell lysates (*n* = 3 independent replicates), using an enzyme-linked immunosorbent assay (ELISA DiaMetra, SpelloItaly) according to the published protocol, with minor changes [23]. Briefly, samples were homogenized in RIPA buffer: 50 mM Tris (pH 7.4), 150 mM NaCl, 0.1% SDS, 0.5% Na-deoxycholate, Nonidet P-40, protease, and phosphatase inhibitor mixture (Sigma), using the mortar and pest with liquid nitrogen for the liver samples. After centrifugation (10 min, 5000× *g* at 4 °C), the supernatants were collected and stored at −80 °C until hormones’ measurement was performed by the ELISA kit (Free T3, DKO037, sensitivity 0.05 pg/mL; Free T4, DKO038, sensitivity 0.05 ng/dL) or used for Western blotting analyses.

### 4.6. Reverse Transcription (RT) Quantitative (q) PCR

Total RNA from liver and from HepG2 cells was prepared with TRIzol reagent (Invitrogen, Waltham, MA, USA). RNA (1 µg) was reverse transcribed using a QuantiTect Reverse Transcription Kit (Qiagen, Germantown, MD, USA) and real-time qPCR (RT-qPCR) was performed using PowerUP SYBR Green Master Mix (Applied Biosystems (Foster City, CA, USA) with Applied Biosystem *QuantStudio 7 Flex System*). A total of 10 ng of cDNAs and 300 nM of primers were used, and primer sequences are listed in Appendix A). The reaction was incubated at 95 °C for 10 min, followed by 40 cycles of 95 °C for 15 s and 60 °C for 1 min. Data were normalized by the level of internal control *Beta-actin* expression in each sample. Experiments were performed in triplicates. The 2^−ΔΔCt^ method was used to calculate relative expression changes.

### 4.7. Western Blot Analysis

The liver was crushed using a mortar and pestle, with liquid nitrogen, whereas HepG2 cells were pelleted and washed in PBS. RIPA buffer (50 mM Tris (pH 7.4), 150 mM NaCl, 0.1% SDS, 0.5% Na-deoxycholate, Nonidet P-40, protease, and phosphatase inhibitor mixture (Sigma) was used to prepare whole protein from liver and cells. Upon centrifugation (5 min at 13,000× *g*), the extracted proteins were quantified and used for Western blotting analyses. The primary antibodies’ preparation was performed using antibody solution (TBS 1X, BSA 5% and 0.02% sodium azide). Dilutions of 1:1000 were used for p-FOXO1a Ser256 (ab131339; Abcam, Cambridge, UK), FOXO1a (#2880; Cell Signaling, Danvers, MA, USA), Insulin R-beta (sc-57342; Cell Signaling), Insulin receptor phosphoY972 (ab5678; Abcam), IRS1 tot (#06-248; Millipore, Burlington, MA, USA), p-IRS1Ser302 (#2384; Cell Signaling), AKT(pan) (#4691; Cell Signaling), p-AKT Ser473 (#9271; Cell Signaling), GS3K (pan) (#9315; Cell Signaling), p-GS3KSer21/9 (#9331; Cell Signaling), and DIO3 (NBP 05767; Novus Bio, Littleton, CO, USA) at 4 °C, overnight. A dilution of 1:3000 of B-actin and GAPDH was used as the internal control. The secondary antibodies used were anti-rabbit (G21234, 1:2000) and anti-mouse (G21D40, 1:2000) (Life Technologies, Carlsbad, CA, USA) at room temperature for one hour. Nitrocellulose membranes were stripped two times with NaOH 0.2 M (in water) for 10 min at room temperature and washed two times with water, blocked with milk 2% (in TBS 1X), and reincubated with primary antibodies overnight at 4 °C. Proteins were detected by a chemiluminescence detection method (Millipore, ECL Immunoblot). Signals were quantified by means of a Bio-Rad ChemiDoc™ XRS, using dedicated software (QuantityOne, Bio-Rad Laboratories, Hercules, CA, USA).

### 4.8. Statistical Analysis

A statistical analysis was performed using the Prism 5.0 software (GraphPad Software, La Jolla, CA, USA). One-way ANOVA (post hoc test: Dunnett’s) for multiple comparison was used. Probability *p*-values < 0.05 were considered significant. The results are expressed as the mean ±  standard deviation.

## Figures and Tables

**Figure 1 ijms-24-09582-f001:**
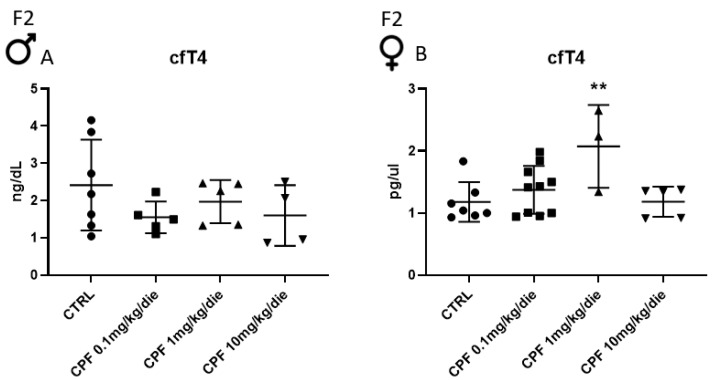
Circulating free levels of T4 (cfT4) in the second-generation (F2) mice exposed to Chlorpyrifos (CPF). The cfT4 serum content in 6-month F2 males (at least 3 mice) (**A**) and females (at least 3 mice) (**B**) measured by ELISA assay. Serum was prepared and tested, as reported in the Material and Method (M&M) section. Data are reported as mean ± SD. One-way ANOVA, ** *p* < 0.01 vs. CTRL.

**Figure 2 ijms-24-09582-f002:**
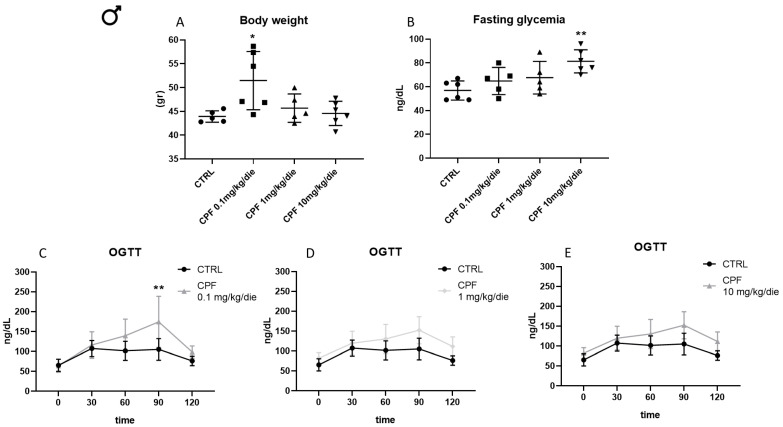
Chlorpyrifos (CPF) induces an increased body weight and fasting hyperglycemia in exposed F2 males. (**A**) Body weight of 6 months old F2 males treated with CPF 0.1, 1, and 10 mg/kg/die (at least 5 animal/group). One-way ANOVA, * *p* < 0.05 vs. CTRL. (**B**) Serum glucose levels determined after 16 h of fasting in mice treated with CPF 0.1, 1, and 10 mg/kg/die. One-way ANOVA, ** *p* < 0.01 vs. CTRL. (**C**–**E**) Plasma glucose concentrations during the oral glucose tolerance test (OGTT) determined at 0, 30, 60, 90, and 120 min in mice treated with CPF 0.1, 1, and 10 mg/kg/die (n = 5). Data are reported as mean ± SD. One-way ANOVA, ** *p* < 0.01 vs. CTRL.

**Figure 3 ijms-24-09582-f003:**
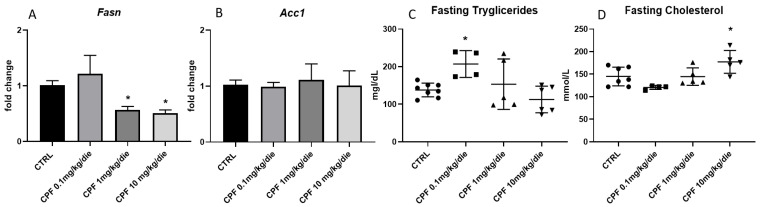
Lipid profiles in Chlorpyrifos (CPF)-exposed F2 male mice. (**A**,**B**) Hepatic levels of *Fasn* and *Acc1* mRNAs determined by RT-qPCR. Data were normalized to the values obtained for CTRL animals (set as 1) and shown as mean ± SD. We used at least 3 samples for the analysis. One-way ANOVA, * *p* < 0.05 vs. CTRL. (**C**,**D**) Serum content of triglycerides and cholesterol were measured by ELISA assay (at least 4 animals). Data are reported as mean ± SD. Data were obtained considering at least 4 samples. One-way ANOVA, * *p* < 0.05 vs. CTRL.

**Figure 4 ijms-24-09582-f004:**
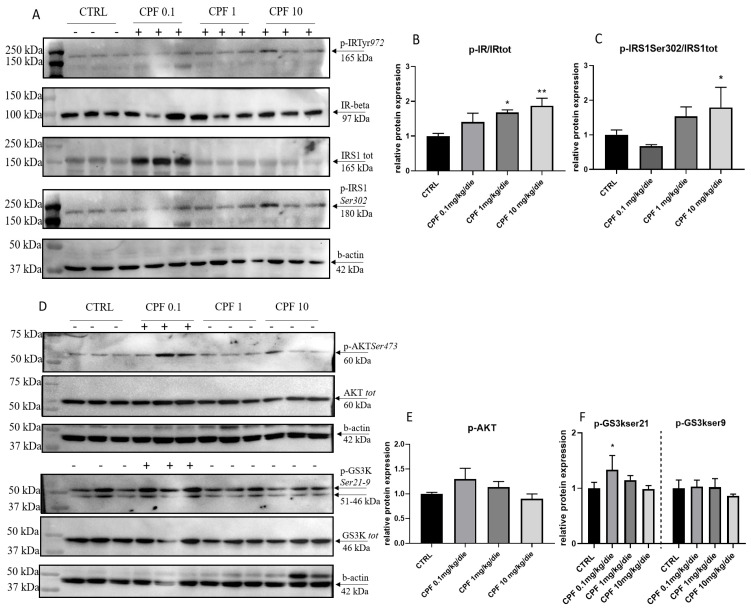
Modulation of the insulin signaling pathway in Chlorpyrifos (CPF)-exposed F2 males. (**A**,**D**) Representative Western blot of proteins prepared from 6-month F2 liver probed with different antibodies against pIRTyr972 (**B**), p-IRS1Ser302 (**C**), p-AKTSer473 (**E**), and pGS3KSer9/21 (**F**). Data were normalized for b-actin protein. Western blots were quantified by densitometry, using Image Lab Software. Results were obtained from three independent experiments and expressed as mean and ± SD. One-way ANOVA, * *p* < 0.05 and ** *p* < 0.01 vs. CTRL.

**Figure 5 ijms-24-09582-f005:**
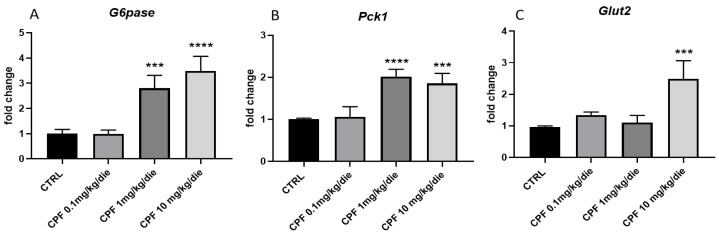
Glucose metabolic profiles in Chlorpyriphos (CPF)-exposed F2 males. Hepatic levels of *G6pase* (**A**), *Pck1* (**B**), and *Glut2* (**C**) mRNAs were investigated by RT-qPCR. Data were normalized to the values obtained for CTRL animals (set as 1) and shown as mean ± SD. At least 3 samples were used in the analysis. One-way ANOVA, *** *p* < 0.001, and **** *p* < 0.0001 vs. CTRL.

**Figure 6 ijms-24-09582-f006:**
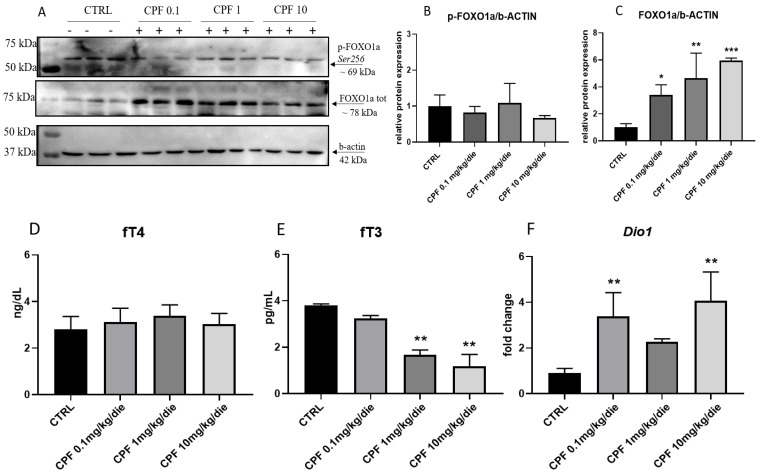
Intra-hepatic levels of FOXO1A protein and of thyroid hormones (THs) in F2 livers exposed to Chlorpyrifos (CPF). (**A**) Representative Western blotting analysis of the liver extracts of the exposed F2 males probed with p-FOXO1ASer256 and FOXO1a total form. Data were normalized for b-actin protein. (**B**,**C**) Western blots were quantified by densitometry, using Image Lab Software. Results were obtained from three independent experiments and expressed as mean and ± SD. One-way ANOVA, * *p* < 0.05, ** *p* < 0.01, and *** *p* < 0.001 vs. CTRL. Intra-hepatic content of free Thyroxine, T4 (fT4) (**D**), and free triiodothyronine, T3 (fT3) (**E**), in F2 males treated with CPF 0.1, 1, and 10 mg/kg/die determined by ELISA assay in tissue homogenates (N = 3). One-way ANOVA, ** *p* < 0.01 vs. CTRL. (**F**) Expression levels of hepatic deiodinase type 1 (*Dio1*) transcript measured by RT-qPCR. Data were normalized to the values obtained for CTRL animals (set as 1) and shown as mean ± SD. At least 3 samples/group. One-way ANOVA, ** *p* < 0.01 vs. CTRL.

**Figure 7 ijms-24-09582-f007:**
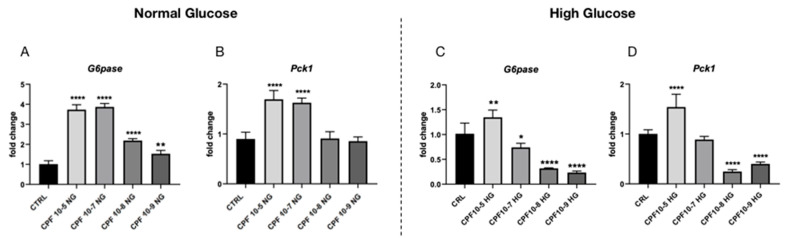
A higher dose of Chlorpyrifos (CPF) stimulated glycolysis and gluconeogenesis processes in human hepatic carcinoma cells line, HepG2, grown in normal- and high-glucose medium, respectively. mRNA levels of *G6pase* (**A**,**C**) and *Pck1* (**B**,**D**) were determined by RT-qPCR. Data were normalized to the values obtained for CTRL (set as 1) and shown as mean ± SD (three independent replicates). One-way ANOVA, * *p* < 0.05, ** *p* < 0.01, and **** *p* < 0.0001.

**Figure 8 ijms-24-09582-f008:**
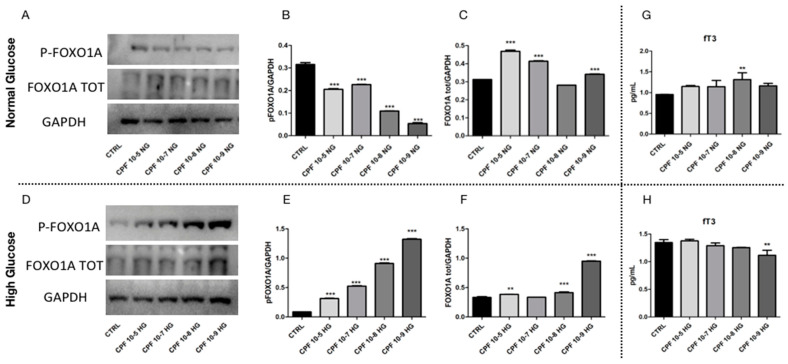
Chlorpyriphos (CPF) effects on FOXO1-T3 axis in human hepatic carcinoma cells line and HepG2 cultured in normal- and high-glucose condition. (**A**–**F**) Representative Western blot of extracts from HepG2 cells cultured in both normal- and high-glucose condition (NG (**A**–**C**) and HG (**D**–**F**), respectively) and exposed to different doses of CPF (as detailed in Section 4), probed with p-FOXO1Aser256 and FOXO1A. Data were normalized for GAPDH protein. Western blots were quantified by densitometry using Image Lab Software. Results were obtained from three independent experiments and expressed as mean and ± SD. One-way ANOVA, ** *p* < 0.01, and *** *p* < 0.001 vs. CTRL. ELISA assay for fT3 in HepG2 extract cultured in NG (**G**) and HG (**H**) medium and treated with different CPF doses. Data represent the mean and the SD. Data represent three independent biological replicates. One-way ANOVA, ** *p* < 0.01.

## Data Availability

The data presented in this study are available within this article and Appendix A.

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
