# Peer review of "Chronic Exposure to Chlorpyrifos Damages Thyroid Activity and Imbalances Hepatic Thyroid Hormones Signaling and Glucose Metabolism: Dependency of T3-FOXO1 Axis by Hyperglycemia"

_ijms, 2023, doi:10.3390/ijms24119582_

Round 1

Reviewer 1 Report (New Reviewer)

There is an extensive literature on chlorinated organophosphate toxicity.  The authors have addressed two aspects, one glucose metabolism related, the other thyroid.   It is difficult to jump back and forth between the two aspects.

The experiments are fine and nicely described.  What is lacking is a sense of relevance to real life toxicity.  We need correlation to what has already been done by others.  

Here is a review 

Ubaid Ur Rahman H, Asghar W, Nazir W, Sandhu MA, Ahmed A, Khalid N. A comprehensive review on chlorpyrifos toxicity with special reference to endocrine disruption: Evidence of mechanisms, exposures and mitigation strategies. Sci Total Environ. 2021 Feb 10;755(Pt 2):142649.

Can the authors broaden their literature search to convince the reader that the exposures they carried out experimentally are not far in excess of what could be expected from contamination  by pesticides.   If the concentrations are far beyond what could happen, that is perfectly all right to confirm relative safety.  

It would also be helpful to separate this manuscript into two sections, one glucose related, the other thryoid.

Author Response

Reviewer 2 Report (New Reviewer)

This is an interesting study describing metabolic changes induced in the offspring of mice treated with CPF via thyroid dysregulation. Although the presented data are valuable and the authors did a thorough job investigating the subject, the manuscript itself is badly organized and difficult to follow. Hence, I suggest the authors to rewrite parts of the manuscript before resubmission.

Major concerns:

1. The Abstract should explain in a few words what chlorpyrifos is. The explanation of abbreviation TH is also lacking in the abstract.

2. The Introduction section and the Abstract do not mention (let alone explain) the vast majority of mRNAs investigated in the Results section (Dio1, Fasn, Acc1, G6pase, Pck1 etc…). The Introduction section only gave enough information to be able to follow the first two figures.

3. All figures need titles

4. In result analysis, the authors used One-way ANOVA to check for between-group differences. However, Pearson’s correlation (or Spearman’s where appropriate) should also be employed to check fore dose dependency

Author Response

Reviewer 3 Report (New Reviewer)

The study by Peluso and coworkers presents the mechanism of chlorpyrifos (CPF) action on hepatic homeostasis of thyroid hormones and glucose metabolism. Unfortunately, the manuscript cannot be accepted in the current form. Suggested changes are listed below.

1.       Please explain all the abbreviation at the first mention in the main body of the manuscript (e.g., F1 in line 87, ). Similarly, explain abbreviations in figure legends at the first mention (e.g., ‘CPF’, ‘cFT4’ and ‘F2’ in Fig. 1 legend or ‘OGTT’ in Fig. 2 legend).

2.       I would suggest writing gene symbols with all letters in uppercase. It seems to me that it is worth giving two names of the same gene if they are in common use (e.g., ‘GLUT2 (SLC2A2)’).

3.       I would advise you to avoid serif fonts in figures (e.g. Figs. 6A, 7A and D).

4.       Please standardize the spelling of amino acid names. Sometimes one-letter notations are used (e.g., in Fig. 7A ‘Y’ as tyrosine) and sometimes three-letter abbreviations (e.g., and in Fig. 7A ‘Ser’ as serine).

5.       Line 309: References are needed.

6.       Manufacturers of reagents used in the study should be provided (e.g., FBS, DMEM media, antibiotics for culture media).

7.       The Authors use as a decimal separator a period (e.g., in line 147) and a comma (e.g., in line 416). It is worth standardizing the way decimal numbers are written.

8.       Line 442:  I recommend using 'x g' in place of ‘rpm’ when giving centrifugation speed.

9.       Materials and Methods (RT-qPCR): Please provide more details regarding (i) primers concentration, (ii) dilution factor for cDNA used in RT-qPCR mixture, (iii) cycling conditions.

10.   Materials and Methods (Western blot): Please add (i) composition of washing buffer, blocking buffer, diluent for primary and secondary antibodies, (ii) times and temperatures of incubation with blocking agent and antibodies, (iii) chemiluminescent substrate details, (iv) stripping conditions for reprobing.

Examples of typos:

1.       Please standardize the notation ‘vs’. This notation appears in various forms (e.g. italics or not, with or without a dot).

2.       Line 109: Change ‘CPF exposed’ for ‘CPF-exposed’.

3.       Lines 110 and 354: Change  ‘mg/Kg/die’ for ‘mg/kg/die’.

4.       Line 117: Change ‘impat’ for ‘impact.

5.       Line 141: Change ‘dose -dependent’ for ‘dose-dependent’.

6.       Line 206: Change ‘its’ for “it’.

7.       Lines 253, 255 and 261: Change ‘figure’ for ‘Figure’.

8.       Line 412: Change ‘as previous describe’ for ‘as previously described’.

9.       Line 425: Change ‘CPF Doses’ for ‘CPF doses’.

Round 2

Reviewer 1 Report (New Reviewer)

The authors have improved organization.  I am having difficulty understanding the contrast between "developmental" and "lifelong exposure"   do the authors have contrast data between mice exposed developmentally plus lifelong to mice only exposed developmentally?

Author Response

Reviewer 3 Report (New Reviewer)

The manuscript has been improved. However, it is still cannot recommend it for publication.

Major remarks:

1.       Line 105: Please check if expression “Figure S1C e D” is correct. I'm not sure if this figure numbering is correct.

2.       Supplementary Materials: I would suggest changing “Supplementary 1A” for “Supplementary Figure 1A” (or something similar; it is important that the word "Figure" appears). I would suggest a similar change for the rest of the Figures in this supplementary file.

3.       Table 1 in Supplementary Materials: Table legend is needed. Additionally, it is not clear what means “mm”. Please explain.

4.       Line 526: Pease specify how long the membranes were incubated with 0.2 N NaOH, at what temperature the stripping was carried out and how long it lasted. Were the membranes then blocked again with a blocking agent? What was NaOH dissolved in (water, buffered Tris?). I am surprised by the lack of detergent (Tween 20) in the antibody incubation buffer. Please specify the composition of the wash buffer. I assume that the detergent has already been added there (please specify the concentration).

5.       Lines 528-529: Please specify what software was used for the statistical analysis.

Typos (examples):

1.       Lines 86, 176, etc.: Change “RTqPCR” for “RT-qPCR”.

2.       Line 94: Change “to Chlorpyrifos (CPF) of” for “to CPF of”.

3.       Line 100: Change “Fig S1A” for “Fig. S1A”.

4.       Figure 1: Change “Kg” for “kg”. Similar change is suggested in Figure 2, 4, 6 and in Supplementary Materials.

5.       Line 110: Correct “0.1 - 1 and -10 mg/kg/die”.

6.       Lines 123 and 175: Change “vs.” for “vs”. Please use italics.

7.       Line 166: Change “1mg/kg/die” and  “0.1mg/kg/die” for “1 mg/kg/die” and “0.1 mg/kg/die”, respectively.

8.       Line 232, 239, 240, etc.: Change “western” for “Western”.

9.       Line 393, 403 and Supplementary Materials: Change “0,1” for “0.1”. Please correct the spelling of numbers throughout the manuscript.

10.   Figures 6A and 7A: Please change serif font for sans serif font.  

11.   Line 264: Change “pIRtyr972” for “pIRTyr972”.

12.   Lines 314, 317 and 428: Change “figure” for “Figure”.

13.   Line 455: Change “describe” for “described”.

14. Lines 516-526: Please write the names of antibody manufacturers correctly, For example, change “abcam” for “Abcam”.

Round 3

Reviewer 1 Report (New Reviewer)

This is long and difficult paper.    The underlying hypothesis is that chlorpyrifos exposure alters gluconeogenesis by an effect on thyroid hormone metabolism. The data in support is tenuous, given that there is a sex difference and in some cases low doses of chlorpyrifos show effects whereas in other experiments there is a clear dose response.   There is a large amount of material here, many experiments.  The authors would do far better to present their data in several parts, in separate but connected manuscripts.  What we see is a questionable thyroid hormone mediated effect but  a stronger gluconeogenesis effect,   It might be best for the authors to present their data in this way rather than to try to force their concept of thyroid hormone driven effects.  A series of papers would be far more impressive than one massive paper that they have submitted.

SPECIFIC COMMENTS

We determined the 101 circulating level of free thyroxine (T4) in exposed males of the same litter by ELISA 102 assay (cfT4, from now). None statistical difference was detected as in females NO STATISTICAL

CPF produced a more pronOunced activity of FOXO1a protein, even without a strong 319 impact on T3 levels. 320

Conversely, in HG, CPF generated a statistically significant reduction

This might depend on the time anyway required to develop the 376 phenotype due out long lifespans and the relative smaller number of offspring in hu-377 man. Nevertheless, our results suggest a risk for developmentally exposed individu-378 als of developing endocrine/metabolic diseases later in the life or subsequently to ex-379 posure to other environmental cues. These lasts might be represented by the alteration 380 of other signaling pathways linked or not to the exposure. It is well known that THs 381 regulates glucose metabolism and homeostasis THIS IS HARD TO FOLLOW

This manuscript is a resubmission of an earlier submission. The following is a list of the peer review reports and author responses from that submission.

Round 1

Reviewer 1 Report

This manuscript investigates the effect of CPF exposure on thyroid activity and hepatic glucose homeostasis in F2 males. This topic is important and interesting; however, I have found several serious concerns that dampen my enthusiasm for this manuscript.

I did not see any piece of evidence showing the connection of T3 with hepatic changes observed in F2 male mice. One can say that dysregulation of T3 and hepatic damage could be independently targeted by CPF in male mice. In other words, the authors did not convince me that CPF-induced dysregulation of T3 leads to changes in Foxo1 and liver impairment/hyperglycemia. Therefore, the results of this manuscript do not support the main conclusion of the existence of the T3-Foxo1 axis.

The authors showed that the level of T3 and DIO-3 in the liver decreased after CPF exposure. DIO-3 inactivates T3 by converting it into T2. If the level of DIO-3 is decreased, I would expect that the level of T3 increases in the liver of F2 males due to the inability of the liver to convert it to T2. However, this is not the case and F2 males show a low level of hepatic T3. 

One of the main points in the manuscript is the protein FOXO1. When I look at Figure 6, the phosphorylation of this protein was not changed after the CPF exposure. Therefore, one can argue that FOXO1 was not involved in the CPF action in F2 males. However, one of the major conclusions in this manuscript is based on the premise that CPF changes the liver FOXO1 activity.  

There is also a problem with signaling results in this manuscript.  CPF at 0.1 mg/kg increases the activation of AKT but this does it does not change the activation of IR. On the other hand, 1 and 10 mg/kg CPF activate IR which does not lead to phosphorylation of AKT. AKT is one of the main targets of IR and it is strange that activation of IR does not phosphorylate AKT.

In that line, I noticed an inconsistency regarding the effect of different CPF doses. Besides IR-pAKT signaling, the authors showed that a higher dose of CPF leads to changes in fasting glucose, however, only a dose of 0.1 mg/kg causes changes in OGTT. Then, 1 and 10 mg/kg reduces Fasn, however, 0.1 mg/kg increases fasting triglycerides. Because of that, the result in this manuscript is very difficult to follow. It also put some serious doubts about the toxic level of CPF.

I would also like to draw attention to several inaccuracies in the manuscript:

page 3, line 86, "a trend towards the decrease".... it is not a trend, but a statistically significant decrease

page 3, the title of Figure 1 does not correspond to the results. It is written male mice, however, Figure 1 contains data on male and female mice.

page 4; lines 107-108 "increase in glucose level at 90 min after administration of ... higher dose" is not supported by the data in Fig 2C.

page 6, lines 165 "increase in the levels of cholesterol at the high doses of CPF, is not supported by Fig 4D.

page 9, line 304 "FOXO1 phosphorylation was reduced in exposed F2 male livers.... this is not supported by the results in Figure 6.

Reviewer 2 Report

How do the authors explain normal T4 with high D1 and low D3 (lines 135-140)? And there is something wrong with the dose-dependent results to D1

Which is the CPF dose that is expected to be found in humans? 

Lines 295-298 - hard to explain this combination, please develop more insights. Low D1 and low T3? How?

Authors should explain in more detail the results with the mechanism of gluconeogenesis. 

Reviewer 3 Report

Authors investigated the impairment of metabolism/signaling of THs and glucose metabolism in mice developmentally and lifelong exposed to 0.1-,1- and 10 mg/kg/die CPF (F1 generation) and their offspring similarly exposed (F2 generation). 

Authors particularly pointed to peripheral disarrangement of thyroid hormone(s) signaling and its crosstalk with around 70 other pathways, involved in hepatic control of glucose metabolism, as the mechanism of chlorpyriphos (CPF) action.

I would suggest that authors' assessment of the effects (dose-response relationship) using PROAst software and estimating BMDL (in % that is significant for each measured parameter. Only in the case if effects are changed in a dose-dependent manner that could be described by software authors can discuss effects (please check EFSA Benchmark dose support for use from 2017 and 2014).

Of particular interest is that the authors chose the doses based on  published reports in which not systemic effects of the 368 exposure were reported consistent with the NOAEL for CPF (0.3 mg/kg/die) and its 369 relevant long-term NOAEL (0.9 mg/kg/die, 18 months) reported in the last statement 370 of EFSA for CPF (approved 31 July 2019). After calculation of BMDL (Benchmark dose lower confidence limit there is excellent option for comparison with NOAELs and make comments for previously assessed risk, is it over or underestimated.

The publication is based on plenty of highly valuable results.

Maybe comparison with the OECD guideline or test method of Regulation 440/2008 EC can contribute to the data relevance.